# Plants and Mushrooms as Possible New Sources of H_2_S Releasing Sulfur Compounds

**DOI:** 10.3390/ijms241511886

**Published:** 2023-07-25

**Authors:** Valentina Citi, Marco Passerini, Vincenzo Calderone, Lara Testai

**Affiliations:** 1Department of Pharmacy, University of Pisa, Via Bonanno, 56120 Pisa, Italy; valentina.citi@unipi.it (V.C.); vincenzo.calderone@unipi.it (V.C.); 2Golden Wave srl, Via Don G. Lago, 35013 Padova, Italy; marco@goldenwave.pro; 3Interdepartmental Center of Nutrafood, University of Pisa, Via Del Borghetto 80, 56124 Pisa, Italy

**Keywords:** organosulfurs, hydrogen sulfide, *Alliaceae*, *Brassicaceae*, fungi, H_2_S-donors

## Abstract

Hydrogen sulfide (H_2_S), known for many decades exclusively for its toxicity and the smell of rotten eggs, has been re-discovered for its pleiotropic effects at the cardiovascular and non-cardiovascular level. Therefore, great attention is being paid to the discovery of molecules able to release H_2_S in a smart manner, i.e., slowly and for a long time, thus ensuring the maintenance of its physiological levels and preventing “H_2_S-poor” diseases. Despite the development of numerous synthetically derived molecules, the observation that plants containing sulfur compounds share the same pharmacological properties as H_2_S led to the characterization of naturally derived compounds as H_2_S donors. In this regard, polysulfuric compounds occurring in plants belonging to the *Alliaceae* family were the first characterized as H_2_S donors, followed by isothiocyanates derived from vegetables belonging to the *Brassicaceae* family, and this led us to consider these plants as nutraceutical tools and their daily consumption has been demonstrated to prevent the onset of several diseases. Interestingly, sulfur compounds are also contained in many fungi. In this review, we speculate about the possibility that they may be novel sources of H_2_S-donors, furnishing new data on the release of H_2_S from several selected extracts from fungi.

## 1. Introduction

Hydrogen sulfide (H_2_S) has been known for many decades for its toxicity and the smell of rotten eggs. Hydrogen sulfide poisoning usually occurs by inhalation; local irritant effects result in direct irritation to the eyes, causing conjunctival injection initially and corneal injury eventually. When inhaled, it leads to pulmonary injury presenting as hemorrhagic pulmonary edema. Hydrogen sulfide inhibits mitochondrial cytochrome oxidase by making a complex bond to the ferric moiety of the protein, therefore arresting aerobic metabolism. Once it enters the bloodstream and passes the blood–brain barrier, neurotoxic effects can be seen, namely dizziness, seizure, coma, and ultimately death. In brief, hydrogen sulfide is a known pulmonary irritant and asphyxiant that primarily causes respiratory and neurological clinical manifestations when inhaled. High concentrations (more than 700 ppm) have the potential to cause sudden death [1,2].

However, in the last twenty years, it has been discovered as the third endogenously produced gasotransmitter, in addition to nitric oxide and carbon monoxide [3].

The importance of characterizing and discovering new sources to be used as supplements of H_2_S or developing new chemical entities endowed with H_2_S-donor properties is due to the pleiotropic role of this gaseous molecule. Indeed, H_2_S is involved in the regulation of numerous pathophysiological functions: it intervenes in the anti-inflammatory process [4], in oxidative stress [5], neuro-modulation [6], vaso-regulation [7], protection from ischaemia/reperfusion damage after myocardial infarction [8] and insulin resistance [9]. Furthermore, recent studies have shown that an abnormal metabolism and altered pathways of H_2_S are related to the development of various cardiovascular and non-cardiovascular diseases (Figure 1) [10,11,12].

Numerous synthetically derived molecules have been developed, and the discovery of new interesting scaffolds able to release H_2_S in biological environments, including thioureas [13], isothiocyanates [14,15,16], thiazolydindione [17], arylthioamide [18], and imminothioether derivatives [19], have highlighted the concrete usefulness of using these molecules for the treatment of several pathological conditions. Moreover, H_2_S-donor moieties have also been used for developing hybrid drugs upon merging the chemical portion able to release H_2_S to drugs already used in the clinic [20,21,22,23,24,25,26,27]. This strategy allows us to exploit the beneficial effect of H_2_S and the well characterized pharmacological effect of the native drug.

In addition to synthetic molecules, the observation that plants containing sulfur compounds share the same pharmacological properties with H_2_S led to characterize naturally derived compounds as H_2_S donors. Polysulfuric compounds occurring in plants belonging to the *Alliaceae* family were the first to be characterized as H_2_S donors, followed by isothiocyanates derived from vegetables belonging to the *Brassicaceae* family [28,29]. This important discovery paved the way to consider these plants as nutraceutical tools and their daily consumption has been demonstrated to prevent the onset of several diseases [30]. Sulfur compounds are also contained in many fungi. However, to date no data about their possible H_2_S releasing properties have been reported.

This review aims to describe the pharmacological effects of sulfur compounds present in plants belonging to the *Alliaceae* and *Brassicaceae* families and provide new data on the H_2_S-releasing properties of fungi extracts.

## 2. Hydrogen Sulfide Endogenous Production

In mammals, the endogenous production of this gasotransmitter is ensured by several enzymes, that are differently expressed in tissues, suggesting its crucial role in the control of homeostasis in numerous systems. Recently, H_2_S has been reported to modulate target proteins through S-sulfhydration reactions, which are a post-translational modification of the hydrosulfuryl groups of cysteine residues responsible for the activation/inhibition of specific proteins [31,32].

H_2_S is mainly bio-synthetized using two cytosolic enzymes: cystathionine γ-lyase (CSE) and cystathionine β-synthase (CBS), and one mitochondrial enzyme: 3-mercaptopyruvate sulfur-transferase (MST), using a shared substrate, the amino acid L-cysteine [33]. Interestingly, CBS is mainly but not exclusively expressed in the central nervous system (CNS), while CSE is recognized as the enzyme responsible for the endogenous production of H_2_S in the cardiovascular system. More recently, additional/alternative biosynthetic pathways have been described; indeed, H_2_S can be generated by cysteinyl-tRNA-synthetase enzyme and by selenium-binding protein 1, although their contribution is still unclear [34]. The wide distribution of biosynthetic pathways for the endogenous production of H_2_S accounts for its ubiquitous and pleiotropic behavior.

With regards to metabolic processes, H_2_S can follow different pathways: firstly, H_2_S is a reducing molecule, and it is involved in reducing reactive oxygen species (ROS). Oxidation of H_2_S represents the main catabolic function at the mitochondrial level, where H_2_S is quickly metabolized to sulfite and sulfate species due to the involvement of quinone oxidoreductase, rhodanese, and sulfur dioxygenase [32].

## 3. Overview of Hydrogen Sulfide and Its Therapeutic Potential in Oxidative Stress-Based Pathologies

### 3.1. Oxidative Stress

It is well-known that oxygen is an indispensable molecule, but because of its structure, it can lead to the generation of highly unstable intermediates known as ROS [35]. When the ROS level exceeds the neutralizing capability of endogenous anti-oxidant systems, the tissue is exposed to oxidative stress. The accumulation of ROS is implicated in mitochondrial dysfunction of the cells due to oxidation of mitochondrial lipids leading to a decrease in the membrane potential and finally apoptosis [36]. In this regard, the nuclear transcription factor erythroid 2 (Nrf-2) is the main regulator of the anti-oxidant defense mechanisms within the cell. Nrf-2 modulates the gene expression of anti-oxidant and cytoprotective enzymes and is inhibited by Kelch-like ECH-associated protein 1 (Keap1). For this reason, Nrf-2 represents a possible promising pharmacological target to decrease cellular oxidative stress [37].

Although many studies have demonstrated direct H_2_S scavenging action, the physiological concentration of H_2_S is in the low nanomolar range, significantly lower than that of other endogenous anti-oxidant molecules [38]. Consequently, the anti-oxidant effects described for H_2_S comprise the up-regulation of anti-oxidant defense systems [39].

H_2_S is known for its ability to protect from oxidative stress by inducing an S-sulfhydration reaction of the Keap1 protein, in cysteine-151 residue (Figure 2B). This reaction induces a conformational change in Keap1 and its consequent dissociation from Nrf-2, which translocates into the nucleus. Here, Nrf-2 binds to promoters containing the anti-oxidant response element (ARE) gene sequence, favoring the expression of ARE-dependent genes encoding for major anti-oxidant enzymes, such as glutathione reductase, Catalase (CAT), Superoxide dismutase (SOD), and for non-enzymatic anti-oxidants, such as glutathione (GSH) and Thioredoxin 1 (Trx-1) [40].

### 3.2. Inflammation

H_2_S promotes a significant anti-inflammatory effect mainly by inhibiting the pro-inflammatory transcription factor Nuclear factor kappaB (NF-kB) and preventing mast cell degranulation [41,42]. In a model of induced inflammatory response in pulmonary artery endothelial cells, H_2_S inhibits the inhibitor of κβ kinase (IKκβ) enzyme activity through S-sulfhydration of the Cysteine 179 residue, thereby preventing NF-kB translocation into the nucleus [43,44] (Figure 2A).

H_2_S also plays a regulatory role in macrophages, which are traditionally classified into two subgroups: M1, which are classically activated macrophages, are differentiated by the action of Th1 cytokines, and M2, which are alternatively activated macrophages, are differentiated by the action of Th2 cytokines, such as IL-4 or IL-13 [45].

M1s produce pro-inflammatory cytokines, and are important for antibacterial defense, while M2 macrophages mainly produce anti-inflammatory mediators and are involved in the anti-inflammatory and repair response [46]. H_2_S promotes the polarization of macrophages toward M2, thus involving an anti-inflammatory effect (Figure 2D) [47].

### 3.3. Immune System

H_2_S is also involved in the regulation of the immune system, playing a pivotal role in the regulation of neutrophils. Interestingly, neutrophils express all three major enzymes responsible for H_2_S production, although recent studies have failed to measure detectable activity of CBS in the homogenate of human neutrophils [48]. Even if the mechanisms are complex, research data suggest that H_2_S inhibits the adhesion of neutrophils [48]. The molecular mechanisms by which H_2_S reduces leukocyte migration and adhesion in various experiments are different: activation of different classes of potassium channels [49], induction of heme oxygenase-1 [50], and downregulation of CD11b expression [51]. However, in the presence of a strong pro-inflammatory stimulus, H_2_S stimulates neutrophil tissue adhesion and infiltration [52] through the upregulation of adhesion receptors, such as intercellular adhesion molecule 1 (ICAM-1) and P-selectin [53]. It is not contradictory with data reported earlier: under a septic condition, an increase in leukocyte infiltration is desirable since the organism is compromised.

Studies performed under microbial infection conditions showed that inhibition of H_2_S biosynthesis increased the capacity for bacterial migration (and thus reduced the rate of killing) as fewer mycobacteria were internalized in the acidic vesicles of the macrophages [54]. This suggests that the biosynthesis of H_2_S by the host organism is necessary for proper macrophage activity. In fact, when neutrophils were co-cultured with *E. coli* HB101 in the presence of H_2_S for 24 h, the elimination of bacteria was more efficient than in the absence of H_2_S [32].

### 3.4. Cardiovascular and Metabolic Systems

At the vascular level, H_2_S induces most of its effects by S-sulfhydration of proteins, such as ion channels, enzymes, and receptors, which thereby undergo a conformational change responsible for their activation or inhibition [55,56].

One of the first reported mechanisms of action underlying H_2_S-induced vasodilation consists of the activation of ATP-sensitive potassium (KATP) channels with the subsequent hyperpolarization of vascular smooth muscle cells [57]. Subsequently, it was shown that another important mechanism of action responsible for H_2_S-induced vasodilation is the activation of voltage-dependent potassium channels belonging to the Kv7 family [58]. H_2_S is also able to inhibit 5-phosphodiesterase (5-PDE) enzyme and has also been described as an endothelium-derived hyperpolarizing factor (EDHF) (Figure 2C) [59]. Several studies shed light on the cardioprotection played by this gasotransmitter against several kinds of damage, including the ischemia-reperfusion, diabetic cardiomyopathy, myocardial infarction, and drug-induced cardiotoxicity [60,61]. Although, multiple targets seem to be implicated, among which the regulation of nitric oxide (NO) levels [62], the up-regulation of 5’ adenosine monophosphate-activated protein kinase (AMPK) and Nrf-2 [63], the stimulation of mitochondrial ATP-sensitive potassium (mitoKATP) channels has been probably the most investigated target [64].

Moreover, H_2_S is considered a master regulator of systemic metabolism, indeed through the S-sulfhydration of the insulin receptor substrate 1 (IRS-1), it may maintain insulin sensitivity and, upon direct S-sulfhydration at the cysteine139 site, it may increase peroxisome proliferator-activated receptor γ (PPARγ) activity, thereby changing glucose into triglyceride storage in adipocytes (Figure 2E) [65,66].

Based on the numerous beneficial effects that can be mediated by this gasotransmitter, great attention is being paid to the discovery of molecules able to release H_2_S in a smart manner, i.e., slowly and for a long time, thus ensuring the maintenance of its physiological levels and preventing “H_2_S-poor” diseases.

## 4. Naturally Derived H_2_S-Donors

### 4.1. Polysulfides from the Alliaceae Family

Traditional medicine suggests daily consumption of garlic (*Allium sativum* L.) to maintain the physiological values of systolic and diastolic pressure, and several studies support the effects of garlic extracts and its polysulfides as cardiovascular-protective compounds [67,68,69]. The most abundant polysulfides present in garlic and in other edible species from the *Alliaceae* family, such as chives, onion, and shallot, are allyl sulfide derivatives, including diallyl disulfide and diallyl trisulfide, produced by the cleavage of odorless molecules, including S-alk(en)yl cysteine sulphoxide precursors due to the activity of alliinase enzyme. These secondary metabolites are S-methyl cysteine sulphoxide, S-allyl cysteine sulphoxide (known as alliin), S-propyl cysteine sulphoxide, and S-transprop-1-enyl cysteine sulphoxide [70]. Consequently, after the disruption of cells (upon cutting or cooking), alliinase enzyme, present in vacuoles, it is released and the reaction responsible for the formation of typical polysulfides occurs (Figure 3). The enrichment in organosulfur compounds is closely correlated with the aging of the vegetable; interestingly, it has been reported that reactive organosulfur compounds, such as allicin, are converted to their stable isoforms, such as S-allyl cysteine [71].

H_2_S release is the main mechanism accounting for the vascular effects of polysulfides. The vasorelaxant effects following the administration of garlic extract in rat aortic rings were associated with a release of H_2_S in the isolated organ bath [72]. Finally, the exact mechanism through which H_2_S is released from polysulfides has also been described. They demonstrated that the functional group present in organic thiols, such as L-cysteine or glutathione, can promote a nucleophilic attack on the carbon atom in α-position to the allyl group and then trigger the rupture of the molecule, ultimately favoring the release of H_2_S (Figure 4) [72].

In addition, other beneficial effects of garlic are related to the release of H_2_S, including metabolic ones. Garlic shows potential beneficial effects in T2D [73,74,75]; this vegetable, and its derivatives are endowed with antihyperglycemic effects in genetic animal models of diabetes and in humans, thus preventing cardiovascular complications [76]. Consistently, garlic improves insulin sensitivity and the associated metabolic syndrome in animal models, and its derivatives reduce both insulin resistance and blood glucose in streptozotocin- and alloxan-induced diabetes [77].

Based on the putative targets identified for H_2_S, it has been hypothesized that the contribution of garlic in the maintenance of glucose homeostasis or lipid profile is due—at least in part—to the release of H_2_S, and several putative targets have been identified. Interestingly, polysulfides upregulated and S-sulfhydrated PPARγ and sirtuin 3 (SIRT-3) in cardiomyocytes and such a mechanism was relevant for assuring a prevention of diabetic cardiomyopathy in murine model [78].

### 4.2. Isothiocyanates from the Brassicaceae Family

The discovery about the H_2_S releasing properties of the *Alliaceae* family paved the way for a great breakthrough in the research on plant species containing sulfur compounds, as a nutraceutical approach to be employed at early stages of several diseases. In this scenario, the *Brassicaceae* family is rich in glucosinolates and sulfur compounds that are metabolized into isothiocyanates because of the activation of myrosinase when these plants are cut or shredded [79]. Since there is an overlap between physio/biological effects evoked by isothiocyanates and those exerted by H_2_S (as reported above), the H_2_S-releasing properties of some natural isothiocyanates were evaluated [29]. Most of the compounds generated appreciable release of H_2_S following incubation with cysteine. This behavior is due to the nucleophilic reaction of the thiol group of L-cysteine with the isothiocyanate moiety. This characteristic makes these molecules “smart H_2_S donors” since they can donate the gaseous transmitter only in a biological environment. Recently, the chemical mechanism explaining the generation of H_2_S from isothiocyanates has been clarified: the reaction between L-cysteine and isothiocyanates resulted in the generation of H_2_S due to the generation of dithiocarbamic derivatives between L-cysteine and the isothiocyanate moiety. Intramolecular cyclization leads to the generation of dihydrothiazole derivatives and the formation of H_2_S (Figure 5) [80].

Currently, isothiocyanates are attracting increasing interest mainly for their pharmacological effects at the cardiovascular level. In this regard, a recent study evaluated the vascular effects of erucin related to its H_2_S-donating properties. In this study, the authors demonstrated that erucin releases H_2_S in human aortic smooth muscle cells (HASMCs) in a concentration-dependent manner. Furthermore, a significant antihypertensive effect was reported in spontaneously hypertensive rats after erucin intraperitoneal administration, recording systolic pressure values similar to normotensive rats [81].

Moreover, erucin also promotes protective effects in oxidative stress and LPS-induced damage in endothelial and vascular cells, and at cardiac level [82,83,84].

Finally, an emerging role of H_2_S is the regulation of glucose metabolism: several studies have been performed on *Brassicaceae* as a possible nutraceutical approach in an early stage of diabetes or combination with standard therapy. Consumption of *Alliaceae* and *Brassicaceae* can significantly improve blood glucose control in patients with diabetes, concluding that edible plants that contain sulfur compounds could be useful for treating type 2 diabetes (T2D). Furthermore, supplementation with edible plants containing sulfur compounds could significantly enhance the effect of standard therapy on glucose control [85].

Male balb/c mice fed for 10 weeks with a high-fat diet and supplemented with an extract obtained from the seeds of *Eruca sativa* Mill., rich in glucoerucin, showed a reduction in the body weight gain and improvement of glucose homeostasis. Interestingly, a decrease in white adipose tissue and the size of adipocytes was also observed. In addition, the extract improved adipocyte metabolism by enhancing the activity of citrate synthase and reducing triglyceride levels in mice fed a high-fat diet. Worthy of note, a diet rich in *Brassicaceae* has no adverse effects: the most common adverse events are mild gastrointestinal issues (heartburn, flatulence, stomach discomfort, increased defecation) and hot flashes [85].

Another isothiocyanate capable of releasing H_2_S is moringin. This molecule derives from *Moringa oleifera* Lam., a plant belonging to the *Moringaceae* family that is widely used in the traditional medicine for the treatment of stomach pain, ulcers, vision defects, joint pain and as a digestive [86]. Specifically, the authors analyzed changes in the content of glucosinolates and isothiocyanates in different moringa tissues and measured the H_2_S-releasing properties with the lead acetate assay, a method that takes advantage of the high affinity of divalent lead with H_2_S to form a black precipitate of lead sulfide (PbS). The authors showed that moringa seeds and leaves had a significant amount of total glucosinolates that can be converted to isothiocyanates (mainly benzyl isothiocyanate) by the action of myrosinase [82].

### 4.3. Organosulfur Compounds Present in Mushrooms

Edible mushrooms have been widely used in cooking, but in recent years, their consumption for their medical benefits has increased. In this regard, similar to *Alliaceae* and *Brassicaceae* vegetables, the main areas of use are cardio-metabolic disorders and the regulation of the immune system. A recent systematic review highlights that the consumption of mushrooms—as part of a healthy dietary pattern—is associated with a reduced risk of mortality for cardiovascular diseases, suggesting benefits on cardiovascular and metabolic systems. It has been proposed that these benefits are correlated to hypolipidemic and hypoglycemic effects and anti-oxidants and anti-inflammatory activities [87,88,89,90]. More recently, a putative role in slowing down aging has also been considered and associated with the ability to influence the expression of hallmarks of senescence, including Nrf-2 and FOXOs [91]. Moreover, some authors supposed the role of supplementation with mushrooms to support the treatment of the early stages of neurodegenerative diseases or anti-cancer because of the high anti-oxidant and anti-inflammatory capacities [92,93,94]. Finally, with regard to the regulation of immune system, mushrooms are reported to stimulate cell surface receptor activity, enhancing the activity of natural killer cells, neutrophils and macrophages, responsible for anti-viral and anti-tumor responses [94].

Undoubtedly in these properties, an important contribution is due to the presence of polysaccharides (including β-glucans), enzymes, nucleic acids and other bioactive compounds, including ergosterol and monacolin k [92,93,94]; indeed, usually, the phyto-complex is considered to be responsible for a wide spectrum of beneficial effects; nevertheless today investigations on the mechanisms of action are poor, and a deeper exploration seems to be necessary.

Exploring further bioactive compounds endowed with putative healthy properties, a peculiar characteristic of these vegetables is their distinctive flavor, and the scent is critical in the determination of their quality and popularity. Indeed, a wide variety of volatile organic compounds has been described, and their content, along with non-volatile organic compounds, renders each edible mushroom unique, but at the same time, it can influence medicinal properties. Interestingly, among the volatile organic compounds, organic sulfur compounds have been described [95]. Their recognized precursors are L-cysteine and methionine, which can be converted into thioheterocycles, thioethers, thiols and thiophenes in enzymatic and non-enzymatic ways [96]. A peculiar organic sulfur compound typical of mushrooms and not present in other kinds of vegetable and nonvegetable cells is ergothioneine, an amino acid bearing a cyclized thioureidic group that participates in the health-promoting activity of mushrooms [97].

One of the most studied mushrooms is *Lentinula edodes* (well-known as shiitake). It is rich in sulfurs; indeed, the unique aroma of the mushroom is due to the presence of lenthionine (1,2,3,5,6-pentathiepane), a cyclic sulfur compound [98,99]. In addition to its sensorial properties, lenthionine is endowed with biological effects, including antibiotic and anti-aggregation properties [100]; therefore, great attention has been paid to it.

From a biosynthetic perspective, lenthionine derives from lentinic acid in a two-step enzymatic reaction. In particular, lentinic acid is activated upon the elimination of its γ-glutamyl moiety due to the action of γ-glutamyl transpeptidase, producing an L-cysteine sulfoxide derivative, which then undergoes α,β-elimination, catalyzed by cysteine sulfoxide lyase (better known as alliinase, present in *Alliaceae* vegetables), resulting in a highly reactive sulfenic acid intermediate. The sulfenic acid is then rapidly condensed to form thiosulfinate, which is often further transformed into other sulfur compounds, including lenthionine [101]. Drying conditions, probably due to the Maillard reaction, may deeply influence the organic volatile composition, particularly the content of organic sulfur compounds. The concentration of organic sulfur compounds in shiitake increased in the early stages of drying (about 0.5–1.5 h), and the most representative compounds were dimethyl trisulfide, thioanisole, and lenthionine [102]. In the middle stages of drying the organic sulfur compounds increased with the esters; while in the final stages (4–12 h) their concentration decreased. Interestingly, the sulfur perception as a negative impression of food was hypothesized; nevertheless, it could be interesting from a health point of view [103].

Currently, to the best of our knowledge, no organic sulfur compounds described in shiitake have been described as H_2_S donors; on the other hand, poor information is currently available on the content of organic sulfur compounds in other edible mushrooms. Interestingly, our preliminary studies suggest it is a potentially new source of sulfur compounds endowed with H_2_S-donor properties.

## 5. Analysis of H_2_S Release from Mushroom Extracts

Using an amperometric approach [13,104], we screened a few extracts of mushrooms (listed in Table 1) to explore their ability to release H_2_S.

Surprisingly, the extracts of *Lentinula edodes* Berk., *Ganoderma lucidum* Curtis and *Polyporus umbellatus*, tested using an amperometric method at a concentration of 1 mg/mL in the presence and absence of L-cysteine (4 mM), showed a cysteine-dependent release of H_2_S. In detail, in the presence of this amino acid, *Lentinula edodes* was able to release about 2.5 μM of H_2_S, while *Ganoderma lucidum* released 2.1 μM of H_2_S after 30 min incubation and *Polyporus umbellatus* 2 μM. In the absence of cysteine, there was no increase in the concentration of H_2_S (Table 1 and Figure 6).

Three other extracts showed an H_2_S cysteine-mediated release but reached lower H_2_S levels: *Inonotus obliquus* P. Karst., which showed a release of 1.8 μM, *Agaricus subrufescens* Peck., with a release of 1.6 μM and *Cordyceps sinensis* Berk. with a release of 1.2 μM. For these extracts, the release of H_2_S in the absence of cysteine did not reach the instrument detection threshold (Table 1).

A further group could be identified using extracts with thiol-independent H_2_S-donor properties: *Grifola frondosa* Dicks. which, both in the presence and absence of cysteine, reached values of approximately 1.1 μM and *Hericum erinaceus* Bull. which reached values of approximately 1.2 μM (Table 1).

Finally, the last group could be identified by including those extracts devoid of H_2_S-donor properties, either in the presence or absence of cysteine: the extracts of *Pleurotus ostreatus* Jacq., *Coprinus comatus* O.F. Müll., *Auricularia auricula* Hook. f., *Poria cocos* F.A. Wolf and *Phellinus igniarius* L. did not reach the threshold of instrumental detectability for the release of H_2_S.

Based on quantitative analysis, all mushroom extracts contained significant levels of ergothioneine. *Agaricus subrufescens* Peck., *Pleurotus ostreatus* Jacq., *Coprinus comatus* O.F. Müll., *Lentinula edodes* Berk., and *Grifola frondosa* Dicks. appear to be the extracts with the highest ergothioneine content, far exceeding 200 μg per gram of extract. Nevertheless, the levels of the amino acid detected did not correlate with the release of H_2_S; therefore, we supposed that it did not significantly contribute to this property. Furthermore, using amperometric analysis, commercial ergothioneine incubated at a concentration of 1 mM in the presence and absence of cysteine showed a very modest release of H_2_S, amounting to 0.38 μM. This result suggests that despite the molecule’s chemical–physical potential, it is not capable of donating H_2_S, probably because cyclic thiourea does not exhibit the same reactivity as the previously tested thioureas [13].

The other organosulfur compound previously described in mushrooms is lenthionine. Therefore, we evaluated the release of H_2_S by testing the isolated compound at a concentration of 1 mM. In the presence of cysteine, the release of H_2_S from lenthionine reached very high concentrations (approximately 30 μM after 30 min of incubation) (Table 1). In contrast, in the absence of cysteine, lenthionine showed no release of H_2_S, suggesting that the cysteine-derived thiols can interact with lenthionine, thus leading to the hydrolysis of the molecule and release of H_2_S. Although this compound has already been described in the literature, this is the first time it has been identified as an H_2_S donor. Based on these results, it is possible to hypothesize that *Lentinula edodes* owes H_2_S-donor properties—at least in part—due to the presence of lenthionine. However, to our knowledge, this sulfur compound has not been identified in the other mushroom extracts that emerged from our screening as potential novel sources of H_2_S; therefore, it might be challenging to further characterize the content of organosulfur compounds in mushroom extracts that have been tested.

## 6. Conclusions

*Alliaceae* and *Brassicaceae* families were confirmed to be the main natural sources of organosulfur compounds, endowed with pleiotropic beneficial effects that reflect—under numerous aspects—those exerted by H_2_S.

In addition, among the vegetables in the *Alliaceae* and *Brassicaceae* families, mushrooms emerged as a new putative source of organosulfur compounds endowed with H_2_S-donor properties, and this hypothesis paves the way for a re-analysis of the health benefits of these vegetables and for the exploration of a new field of study in which the gasotransmitter is implicated.

These original data presented in this review represent a novelty in the scenario of research on naturally derived H_2_S donors. In this regard, a qualitative and quantitative analysis of the organosulfur compounds contained in the selected mushroom extracts will have to be carried out in order to associate the release of H_2_S with specific chemical entity/ies. It will be the first step in discovering new natural sources of H_2_S and studying further medicinal or nutraceutical applications of these vegetables.

## Figures and Tables

**Figure 1 ijms-24-11886-f001:**
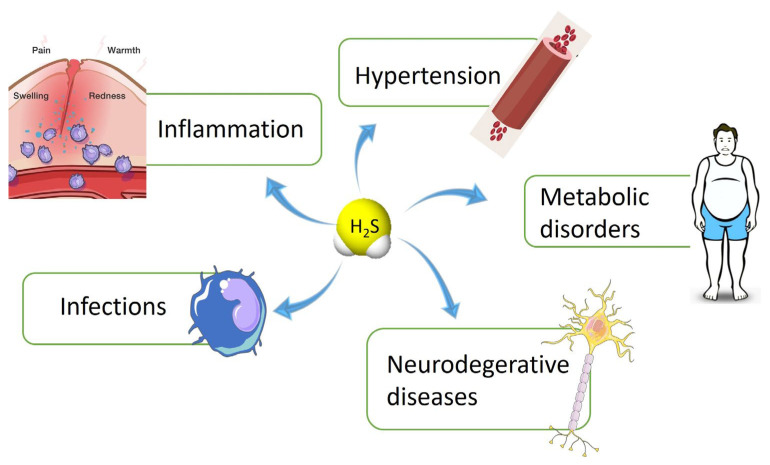
H_2_S-related disorders.

**Figure 2 ijms-24-11886-f002:**
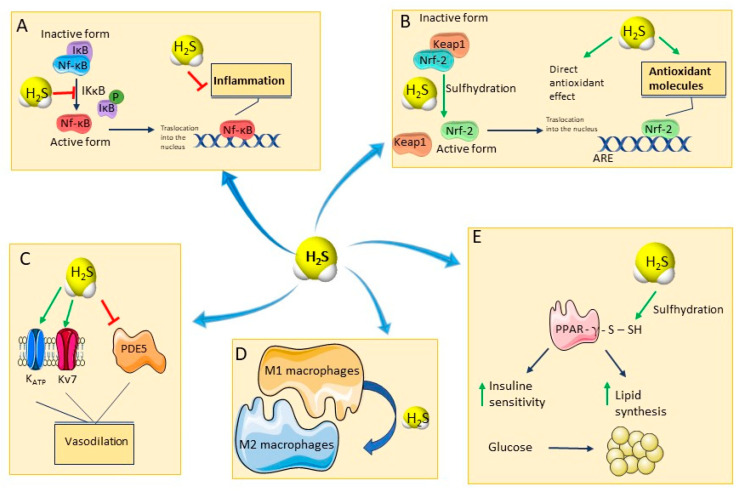
The figure schematically shows the main pharmacological activities of hydrogen sulfide, (**A**) H_2_S and inflammation: through the sulfhydration of IKkB, H_2_S inhibits the dissociation between Nf-kB and IkB, impeding the translocation of Nf-kB into the nucleus and promoting anti-inflammatory effects. (**B**) H_2_S and oxidative stress: H_2_S sulfhydrates Keap1 allowing the dissociation of Nrf-2 which translocates into the nucleus and promotes the expression of anti-oxidant species. Furthermore, H_2_S acts as a direct scavenger of ROS. (**C**) H_2_S and vasodilation: H_2_S induces the opening of potassium channels and inhibits PDE5, leading to hyperpolarization of vascular cells and increase in NO levels. (**D**) H_2_S and immune system: H_2_S facilitates the differentiation in M2 polarized macrophages, which produce anti-inflammatory mediators. (**E**) H_2_S and metabolism: H_2_S sulfhydrates PPAR- γ, increasing insulin sensitivity and lipid synthesis. Abbreviations: Nf-kB: nuclear factor kappa-light-chain-enhancer of activated B cells; IkB: IkappaB kinase; IKkB: inhibitor of nuclear factor kappa-B kinase subunit beta; P: phosphate; Nrf-2: Nuclear factor erythroid 2-related factor 2; Keap1: Kelch-like ECH-associated protein 1; ARE: anti-oxidant responsive elements; KATP: ATP-sensitive potassium channel; Kv7: voltage-gated potassium channels; PDE5: Phosphodiesterase Type 5; PPAR—γ: Peroxisome proliferator-activated receptor gamma.

**Figure 3 ijms-24-11886-f003:**
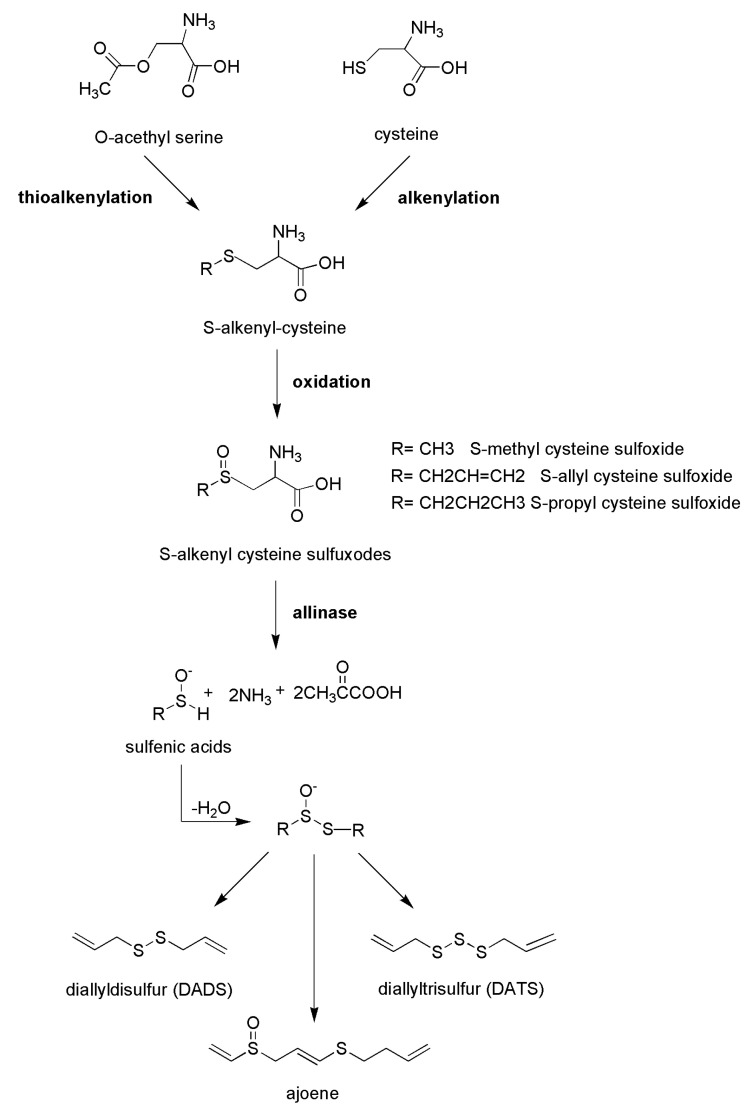
Schematic representation of the biosyntethic reactions leading to the formation of allylsulfur compounds in *Alliaceae* vegetables.

**Figure 4 ijms-24-11886-f004:**
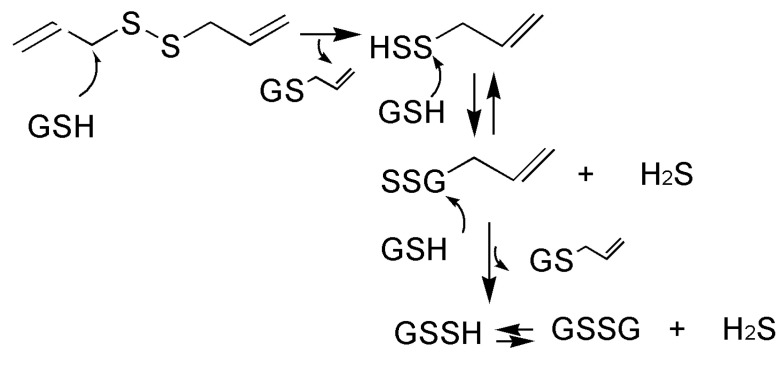
Mechanism of the release of H_2_S from allylsulfur compounds present in *Alliaceae* vegetables. The reaction is triggered by a nucleophilic attack of organic thiols (here represented by glutathione (GSH)) that promotes the formation of H_2_S and perthiols, which generate another molecule of H_2_S and glutathione disulfide (GSSG).

**Figure 5 ijms-24-11886-f005:**
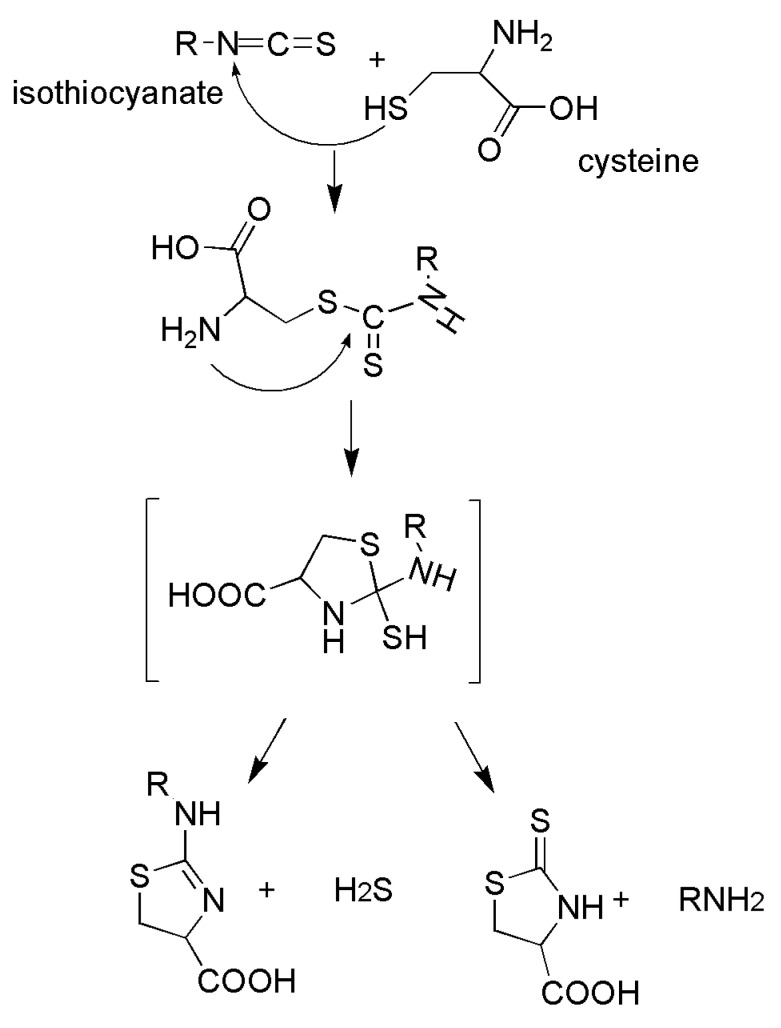
Mechanism of the release of H_2_S from organic isothiocyanates (ITCs) compounds present in *Brassicaceae* vegetables. The reaction is triggered by the nucleophilic attack of thiols (here represented by cysteine) and leads to the formation of an instable intermediate, that spontaneously decomposes in H_2_S and aminic derivatives.

**Figure 6 ijms-24-11886-f006:**
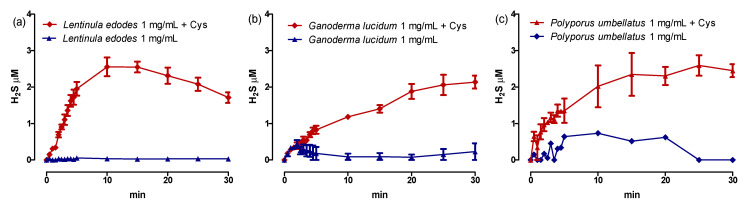
Amperometric recordings of H_2_S release after the incubation of *Lentinula edodes* (**a**), *Ganoderma lucidum* (**b**) and *Polyporus umbellatus* (**c**) 1 mg/mL in the presence or absence of L-Cysteine. Data are expressed as mean ± SEM.

**Table 1 ijms-24-11886-t001:** Maximum H_2_S release after the incubation of the mushroom extracts or ergothioneine and lenthionine.

Tested Item	Ergothioneine Content (µg/g)	H_2_S Release in the Presence of L-Cys 4 mM (μM)	H_2_S Release in the Absence of L-Cys (μM)
*Ganoderma lucidum*	39.1 ± 0.2	2.1 ± 0.2	n.d.
*Hericum erinaceus*	184.6 ± 0.9	1.2 ± 0.2	1.2 ± 0.3
*Lentinula edodes*	348.3 ± 1.2	2.5 ± 0.1	n.d.
*Grifola frondose*	205.1 ± 3.4	1.1 ± 0.2	1.0 ± 0.2
*Polyporus umbellatus*	25.7 ± 0.8	2.0 ± 0.6	n.d.
*Auricularia auricola*	4.24 ± 0.1	n.d.	n.d.
*Inonotus obliquus*	0.3 ± 0.0	1.8 ± 0.1	n.d.
*Agaricus subrufescens*	479.3 ± 4.6	1.6 ± 0.4	n.d.
*Poria cocos*	5.4 ± 0.3	n.d.	n.d.
*Pleurotus ostreatus*	477.7 ± 2.7	n.d.	n.d.
*Coprinus comatus*	450.2 ± 4.8	n.d.	n.d.
*Cordyceps sinensis*	5.9 ± 0.1	1.2 ± 0.4	n.d.
Ergothioneine	n.d.	n.d.	n.d.
Lenthionine	n.d.	32 ± 4.3	n.d.

Abbreviations: L-Cys: L-Cysteine; n.d.: not detected.

## Data Availability

Not applicable.

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
