# Peer review of "Plants and Mushrooms as Possible New Sources of H_2_S Releasing Sulfur Compounds"

_ijms, 2023, doi:10.3390/ijms241511886_

Round 1
Reviewer 1 Report
This review article presents interesting synthesis related to sulfur compounds present in plants as H2S donors and their pharmacological effects with a focus on fungi extracts as a new reported reservoir of H2S-releasing properties. The adopted approach in this study was very consistent since it could provide more insights into the understanding of the pleiotropic role of these molecules.
The manuscript was well introduced, and the authors adopted convincing structure of the topic that pave the way for many promising avenues of research. However, the manuscript needs substantial revisions to be suitable for publication in IJMS.
General comment
Comment 1: The English of this manuscript needs moderate improvements.
Comment 2: A perspectives section is missing.
Comment 3: A section/subsection on H2S toxicity should be developed.
Other comments
The title should be changed to more match the objective of the study at the end of the introduction section.
L36-38: Please provide more references at the end of this sentence since you stated “recent studies” at the beginning.
L42: please delete the coma.
L79: please correct this sentence.
Please specify which subfigure (a, b, c, d or e) each time you cited Figure 1.
L131; please delete the additional space. Please check throughout the manuscript.
L142, 313: please italicize the scientific name.
L193 and 196: please italicize the scientific name.
Figures 2 and 4 please improve the quality.
L212, 237, 242, 248-249, 275, 290, 332, 337: please provide the citation of the reference according to the journal form.
L278: please provide the significance of the abbreviation at the first appearance in the text.
L302: please change “increasing” to “increased”.
L347: please delete “This table summarizes”.
L348: please place the abbreviation significance at the foot of the table.
Table 1: Please change “,” to “.” In the values.
L351: please add a space between the value and the unit. Please check throughout the manuscript.
Table 1 and Figure 5: Please add the reference for both illustrations. If they are your published results please add the reference and if it is not, please mention it.
Figure 5: Please italicize the scientific names.
L358: please remove “This figure represents”.
L401-405: please develop more this section.
The english of the manuscript needs moderate editing.
Author Response
Reviewer 1
This review article presents interesting synthesis related to sulfur compounds present in plants as H2S donors and their pharmacological effects with a focus on fungi extracts as a new reported reservoir of H2S-releasing properties. The adopted approach in this study was very consistent since it could provide more insights into the understanding of the pleiotropic role of these molecules.
The manuscript was well introduced, and the authors adopted convincing structure of the topic that pave the way for many promising avenues of research. However, the manuscript needs substantial revisions to be suitable for publication in IJMS.
General comment
Comment 1: The English of this manuscript needs moderate improvements.
Reply: English has been revised and improved.
Comment 2: A perspectives section is missing.
Reply: Following the suggestion of Reviewer 1, a perspective section has been added in the conclusion section.
Comment 3: A section/subsection on H2S toxicity should be developed. VALE
Reply: Hydrogen sulfide toxicity section has been added in the introduction for a wider point of view on the possible effects of H2S.
Other comments
The title should be changed to more match the objective of the study at the end of the introduction section.
Reply: The title has been changed for better matching the objective of this revie.
L36-38: Please provide more references at the end of this sentence since you stated “recent studies” at the beginning.
Reply: More references have been provided.
L42: please delete the coma.
Reply: Coma has been deleted and other typos have been corrected.
L79: please correct this sentence.
Reply : The sentence has been changed.
Please specify which subfigure (a, b, c, d or e) each time you cited Figure 1.
Reply: Figure 1 subsections (now figure 2) have been specified throughout the manuscript.
L131; please delete the additional space. Please check throughout the manuscript.
Reply: Additional spaces throughout the manuscript have been checked.
L142, 313: please italicize the scientific name.
L193 and 196: please italicize the scientific name.
Reply: The scientific names have been italicized.
Figures 2 and 4 please improve the quality.
Reply: The quality of the figures has been improved (now figure 3, 4 and 5).
L212, 237, 242, 248-249, 275, 290, 332, 337: please provide the citation of the reference according to the journal form.
Reply: The citations have been corrected according to the journal form.
L278: please provide the significance of the abbreviation at the first appearance in the text.
Reply: The significance of the abbreviations has been added.
L302: please change “increasing” to “increased”.
Reply: Increasing has been changed to increased.
L347: please delete “This table summarizes”.
Reply: This statement has been deleted.
L348: please place the abbreviation significance at the foot of the table.
Reply: Abbreviation significance has been moved at the foot of the table.
Table 1: Please change “,” to “.” In the values.
Reply: “,” have been changed to “.”
L351: please add a space between the value and the unit. Please check throughout the manuscript.
Reply: Spaces between the value and the unit have been added throughout the manuscript.
Table 1 and Figure 5: Please add the reference for both illustrations. If they are your published results please add the reference and if it is not, please mention it.
Reply: Table 1 and Figure 5 (now figure 6) refer to unpublished data. This is the first time we provide evidence about the H2S releasing properties of extracts of mushrooms.
Figure 5: Please italicize the scientific names.
Reply: Scientific names in figure 5 (now figure 6) have been italicized.
L358: please remove “This figure represents”.
Reply: The statement has been removed.
L401-405: please develop more this section.
Reply: This section has been improved and developed adding a perspective point of view about founding new H2S vegetable sources.
Comments on the Quality of English Language
The english of the manuscript needs moderate editing.
Reply: English has been improved after reviewing the manuscript.
Reviewer 2 Report
After reading and evaluating the manuscript, I recommend a minor revision to enhance the quality and understandability of the manuscript.
. The Authors should specify the assumptions and novelty of the work.
. Please explain the abbreviation when it is used for the first time.
. species names should be in italics pl revised thoroughly in the entire manuscript.
. All the symbols used in the manuscript, e.g., °C, rpm, etc., should be consistent throughout the entire manuscript.
. Please add and discuss about the recent regulations for H2S (limits and toxicity etc.).
. I suggest incorporating the pictorial representation of the disease associated with the H2S.
. In the title, the authors have focused on the mushrooms than what was the purpose of including the other families like Brassicaceae or Alliaceae family, Sustantiail about of literature and discussion should be done on the mushrooms which are from basidiomycetes broadly and thoroughly discussed. Take different recent past 10-year studies on mushrooms (edible and medicinal) and try to elaborate on their contribution of them as H2S production/donor.
English editing and proofreading should be performed thoroughly before submitting the revised manuscript.
Author Response
Reviewer 2
After reading and evaluating the manuscript, I recommend a minor revision to enhance the quality and understandability of the manuscript.
The Authors should specify the assumptions and novelty of the work.
Reply: The novelty of this work has been specified in the conclusion section and more information about the importance of finding new sources of H2S has been added throughout the manuscript.
Please explain the abbreviation when it is used for the first time.
Reply: The significance of the abbreviations has been added at the first appearance in the text.
Species names should be in italics pl revised thoroughly in the entire manuscript
Reply: The scientific names have been italicized.
All the symbols used in the manuscript, e.g., °C, rpm, etc., should be consistent throughout the entire manuscript.
Reply: The symbols has been corrected to be consistent throughout the manuscript.
Please add and discuss about the recent regulations for H2S (limits and toxicity etc.).
Reply: In the introduction section, a paragraph explaining the limits and toxicity of H2S has been added.
I suggest incorporating the pictorial representation of the disease associated with the H2S. VALE
Reply: Following your suggestion, a figure showing the disease associated with H2S has been added as figure 1.
In the title, the authors have focused on the mushrooms than what was the purpose of including the other families like Brassicaceae or Alliaceae family, Sustantial about of literature and discussion should be done on the mushrooms which are from basidiomycetes broadly and thoroughly discussed. Take different recent past 10-year studies on mushrooms (edible and medicinal) and try to elaborate on their contribution of them as H2S production/donor.
Reply: The paragraph about mushrooms has been developed and improved for a better understanding of the possible applications of the extracts of mushrooms due to their putative H2S donor properties. The discussion about the pharmacological effects of Brassicaceae and Alliaceae has been added to this review for highlighting the importance of finding vegetable sources as H2S donor to be exploited as nutraceutical tools for the prevention/treatment of H2S-deficiency related diseases.
Comments on the Quality of English Language
English editing and proofreading should be performed thoroughly before submitting the revised manuscript.
Reply: English has been improved after reviewing the manuscript.
Reviewer 3 Report
The authors have well written the MS and its new topic of interest especially in the functional/nutraceutical food sector. 1. Is Figure 1 self-designed or borrowed from somewhere else (plz cite the reference)? 2. Quality of figures 2, 3 and 4 is too poor, kindly improve it. 3. Is Table 1 self-designed (plz cite the references) or borrowed from somewhere else (plz cite the reference)? 4. Is Figure 5 self-designed or borrowed from somewhere else (plz cite the reference)?Author Response
Reviewer 3
The authors have well written the MS and its new topic of interest especially in the functional/nutraceutical food sector.
- Is Figure 1 self-designed or borrowed from somewhere else (plz cite the reference)?
- Quality of figures 2, 3 and 4 is too poor, kindly improve it.
- Is Table 1 self-designed (plz cite the references) or borrowed from somewhere else (plz cite the reference)?
- Is Figure 5 self-designed or borrowed from somewhere else (plz cite the reference)?
Reply: Figure 1 (now figure 2) and Figure 5 (now figure 6), and Table 1 are original. Figure 5 and table 1 refer to unpublished data. Quality of Figure 2, 3, and 4 (now figure 3, 4, and 5) has been improved.
Round 2
Reviewer 1 Report
Tyhe authors satisfied all the raised comments. I endorse the publication of the current version of the manuscript.